# The Impact of Smoking-Associated Genetic Variants on Post-Exercise Heart Rate

**DOI:** 10.3390/ijms26188787

**Published:** 2025-09-09

**Authors:** Habib Al Ashkar, Nihad Kharrat Helu, Nora Kovacs, Szilvia Fiatal, Roza Adany, Peter Piko

**Affiliations:** 1Doctoral School of Health Sciences, University of Debrecen, 4032 Debrecen, Hungary; habib.al.ashkar@med.unideb.hu (H.A.A.); nihad.kharrat.helu@mailbox.unideb.hu (N.K.H.); 2HUN-REN-UD Public Health Research Group, Department of Public Health and Epidemiology, Faculty of Medicine, University of Debrecen, 4032 Debrecen, Hungaryadany.roza@med.unideb.hu (R.A.); 3Department of Public Health and Epidemiology, Faculty of Medicine, University of Debrecen, 4032 Debrecen, Hungary; fiatal.szilvia@med.unideb.hu; 4National Laboratory for Health Security, Center for Epidemiology and Surveillance, Semmelweis University, 1089 Budapest, Hungary; 5Department of Preventive Medicine and Public Health, Semmelweis University, 1089 Budapest, Hungary

**Keywords:** smoking, single nucleotide polymorphisms, heart rate recovery, cardiovascular risk, genetic predisposition, YMCA step test, genetic risk score

## Abstract

Smoking has a well-established impact on cardiovascular health, notably through elevated resting heart rate and impaired autonomic regulation—both key risk factors. While nicotine’s acute effects are well documented, the influence of smoking-related genetic variants on heart rate (HR) responses remains unclear. This study investigated the association between selected smoking-related single nucleotide polymorphisms (SNPs) and HR dynamics following physical exertion. A total of 661 Hungarian adults completed the YMCA 3 min step test, with HR measured at rest, immediately post-exercise, and during recovery at 5 and 10 min. Key indices included post-exercise HR (HR_aft_), HR change (ΔHR), maximum HR percentage (HR_max%_), and heart rate recovery coefficient (HRR). Genetic analysis focused on nine SNPs previously linked to smoking behaviours, with a composite genetic risk score derived from the three most influential variants (rs2235186, rs4142041, and rs578776). Associations were examined using adjusted linear regression. No significant relationship was found between any individual SNP and resting HR. However, rs2235186, rs4142041, and rs578776 were consistently associated with elevated HR_aft_, increased ΔHR, higher HR_max%_, and slower HRR. The genetic risk score showed significant correlations with all post-exercise HR measures, suggesting a cumulative genetic effect. These findings indicate that smoking-related genetic predisposition may influence autonomic cardiovascular responses to physical activity. Although resting HR remains unaffected, specific SNPs are linked to post-exercise HR dynamics and recovery, highlighting the potential value of genetic screening in personalised cardiovascular risk assessment among smokers.

## 1. Introduction

Tobacco smoking is a leading global public health concern, responsible for over 8 million deaths annually—primarily due to cardiovascular diseases (CVDs) [1,2,3]. Approximately 22% of adults worldwide smoke, with notable gender disparities: 36.7% of men versus 7.8% of women. Europe has one of the highest smoking rates globally, with an average adult prevalence of 28% [4]. In Hungary, the national rate is similarly high (28%), but among the Roma minority—especially in socioeconomically disadvantaged communities—prevalence exceeds 70% [5]. Early initiation, heavy smoking, and limited access to cessation support contribute to elevated CVD risk in these populations, reflected in Hungary’s persistently high cardiovascular mortality rate [6].

Prolonged smoking induces chronic inflammation; the extra strain imposed on the circulatory system and the reduction in blood oxygen-carrying capacity all contribute to the development of cardiovascular diseases [7]. Nicotine and other tobacco constituents acutely activate the sympathetic nervous system, elevating heart rate (HR) [8], while long-term exposure can disrupt the balance of the autonomic nervous system and cause instability in cardiac rhythm [9].

Beyond resting HR, smoking impairs cardiovascular autonomic regulation, primarily by reducing heart rate variability (HRV)—a key indicator of parasympathetic tone and adaptability. Studies have shown dose-dependent reductions in HRV among smokers [10], acute HR increases after a single cigarette [11], and impaired autonomic reflexes during physiological challenges [12]. Importantly, HRV improves after smoking cessation, indicating that autonomic dysfunction may be reversible [13].

Recent genomic research has begun to explore how single nucleotide polymorphisms (SNPs) associated with smoking behaviour [14,15] and HR regulation [16,17] may influence cardiovascular responses [15,18,19,20]. Several SNPs have been identified as linked to smoking behaviour, nicotine dependence, or dosage intensity [21,22]; it is not yet clear how these genetic variants may influence heart rate regulation, particularly under conditions of physiological stress, such as exercise. Large-scale meta-analyses have identified SNPs that correlate with both smoking behaviour and cardiovascular disease endpoints, suggesting a shared genetic architecture. Twin and family studies further support a heritable component to resting and exercise-induced HR variability, with heritability estimates ranging from 30% to 60% [23,24].

Given the strong epidemiological and physiological links between smoking and HR, and the heritability of these traits, it is plausible that smoking-related genetic variants influence autonomic cardiovascular control beyond behavioural factors. Mendelian randomisation studies have even suggested a causal link between genetic liability to smoking and increased coronary artery disease risk [15,25]. Despite these findings, few studies have directly examined how these variants affect HR responses under physiological stress, such as exercise.

To our knowledge, no prior study has systematically investigated the relationship between smoking-related SNPs and post-exercise heart rate dynamics in a population-based cohort. This study investigates the association between smoking-related SNPs and heart rate dynamics at rest, following physical exertion, and in recovery, to elucidate potential genetic influences on autonomic cardiovascular regulation. Integrating genomic data with physiological phenotyping enhances our understanding of gene–behaviour interactions and may support more precise cardiovascular risk stratification in populations disproportionately affected by smoking.

## 2. Results

### 2.1. Baseline Characteristics and Lipid Profile

A total of 661 individuals were included in the final analysis: 330 non-smokers and 331 smokers, aged 20–64 years. All participants had complete and validated data for assessments of their basic characteristics, genotyping, physical activity, and cardiovascular fitness.

Non-smokers had an average waist circumference of 97.22 cm, while smokers measured 92.71 cm (*p* < 0.001). The mean body mass index (BMI) was 27.96 kg/m^2^ in non-smokers and 26.51 kg/m^2^ in smokers (*p* = 0.001). Systolic blood pressure averaged 126.24 mmHg in non-smokers and 124.24 mmHg in smokers (*p* = 0.048), whereas diastolic blood pressure was 79.90 mmHg and 78.62 mmHg, respectively (*p* = 0.035). Homeostatic Model Assessment of Insulin Resistance (HOMA-IR) values were 4.37 in non-smokers and 3.70 in smokers (*p* = 0.046).

Regarding physical activity, work-related Metabolic Equivalent of Task minutes per week (MET-min/week) values were similar between groups. Transport-related activity was higher among smokers (1669.42 vs. 1345.26 MET-min/week, *p* = 0.007), while leisure-time activity was greater in non-smokers (1340.36 vs. 1007.11 MET-min/week, *p* < 0.001). Sitting time was longer in non-smokers (529.29 vs. 413.44 min/week, *p* = 0.001). Domestic physical activity levels were similar between groups (2930.61 vs. 2976.23 MET-min/week, *p* = 0.790).

Lipid profile analysis showed comparable low-density lipoprotein cholesterol (LDL-C: 3.11 vs. 3.15 mmol/L, *p* = 0.517) and triglyceride levels (1.56 vs. 1.55 mmol/L, *p* = 0.750) between groups. However, high-density lipoprotein cholesterol (HDL-C) levels were lower in smokers (1.26 vs. 1.37 mmol/L, *p* < 0.001).

Heart rate measurements—including resting, post-exercise, and recovery—did not differ significantly between groups. Resting heart rate was 77.20 bpm in non-smokers and 77.61 bpm in smokers (*p* = 0.808). Post-exercise heart rate was 109.79 bpm and 112.25 bpm, respectively (*p* = 0.378). Heart rate recovery at 5 and 10 min, recovery coefficients, and maximum heart rate expressed as a percentage did not differ significantly between groups.

Demographic data indicated that 69.13% of smokers identified as Roma, compared to 32.12% of non-smokers (*p* < 0.001). A higher proportion of smokers reported poor financial status (25.72% vs. 14.85%, *p* < 0.001). Educational attainment differed notably: 71.70% of smokers had completed only primary education or less, while 36.36% of non-smokers fell into this category. College or university degrees were held by 14.85% of non-smokers and 3.54% of smokers (*p* < 0.001).

Alcohol consumption patterns were similar across groups, with no significant differences in frequency. Medication use for hypertension, diabetes, and dyslipidaemia also showed no statistically significant variation between smokers and non-smokers. See Table 1 for more details.

### 2.2. The Best-Fitting Genetic Models by SNPs

For each SNP, the heritable model that showed the strongest correlation with the delta HR was determined. Of the nine SNPs examined, six were found to be dominant; two were found to be recessive, and one was found to be codominant. The strongest correlation was observed with the dominant heritability model for the G allele of rs2235186. See Table 2 for more details.

### 2.3. Association of Smoking-Related SNPs with Resting Heart Rate

None of the nine SNPs examined showed a statistically significant link with resting heart rate (HR_rest_). This finding remained consistent when adjusted linear regression models were used to account for relevant covariates. Figure 1 provides further details regarding these associations, including confidence intervals and the direction of effect for each SNP.

### 2.4. Association of Smoking-Related SNPs with Heart Rate After Exercise

Three SNPs, namely, rs2235186 (B = 3.387, 95% CI: 0.967–5.807, *p* = 0.006), rs4142041 (B = 4.711, 95% CI: 1.990–7.432, *p* = 0.00072), and rs578776 (B = 4.063, 95% CI: 0.873–7.253, *p* = 0.013), showed significant associations with heart rate after physical activity (HR_aft_). See Figure 2 for more details.

### 2.5. Association of Smoking-Related SNPs with Delta Heart Rate

Three SNPs, namely rs2036534 (B = 2.408, 95% CI: 0.390–4.426, *p* = 0.019), rs2235186 (B = 3.508, 95% CI: 1.114–5.902, *p* = 0.004), and rs4142041 (B = 3.980, 95% CI: 1.288–6.672, *p* = 0.004), showed significant associations with delta heart rate (ΔHR). See Figure 3 for more details.

### 2.6. Association of Smoking-Related SNPs with Heart Rate Recovery Coefficient

Two SNPs, namely, rs4142041 (B = 0.041, 95% CI: 0.019–0.063, *p* = 0.00028) and rs578776 (B = 0.030, 95% CI: 0.005–0.05, *p* = 0.021), showed significant associations with the heart rate recovery coefficient (HRR). See Figure 4 for more details.

### 2.7. Association of Smoking-Related SNPs with the Percent of Predicted Maximum Heart Rate

Three of the nine SNPs tested showed a significant association with the percent of predicted maximum heart rate (HR_max%_): rs2235186 (B = 1.848, 95% CI: 0.508–3.188, *p* = 0.007), rs4142041 (B = 2.601, 95% CI: 1.094–4.108, *p* = 0.00075), and rs578776 (B = 2.070, 95% CI: 0.304–3.836, *p* = 0.022). See Figure 5 for more details.

### 2.8. Genetic Risk Score and Its Association with Heart Rate Change in Associated Parameters

Genetic risk score was calculated based on the three SNPs (rs2235186, rs4142041, and rs578776) most closely associated with heart-rate parameters.

Trend analysis revealed a progressive increase in heart rate-related parameters across genetic risk score categories (see Table 3). Although HR_rest_ exhibited no significant trend (*p* = 0.253), HR_aft_, ΔHR, HRR, and HR_max%_ displayed significant positive trends (all *p* < 1 × 10^−6^). Participants with higher genetic risk scores exhibited consistently greater post-exercise heart rate values and enhanced heart rate reactivity.

Complementary regression analysis (Table 4) showed that the GRS was significantly associated with HR_aft_ (B = 3.99; 95% CI: 2.49–5.49; *p* = 2.47 × 10^−7^) and ΔHR (B = 1.64; 95% CI: 0.68–2.60; *p* = 8.86 × 10^−4^), HRR (B = 0.028; 95% CI: 0.016–0.040; *p* = 5.42 × 10^−6^), and HR_max%_ (B = 2.19; 95% CI: 1.36–3.02; *p* = 2.93 × 10^−7^). However, no significant association was observed for HR_rest_ (*p* = 0.343).

## 3. Discussion

This study examined the relationship between nine smoking-related SNPs and heart rate regulation, with a focus on dynamic cardiovascular responses to physical exertion. Although no significant correlations were found with HR_rest_, three variants (rs2235186, rs4142041, and rs578776) demonstrated statistically significant associations with post-exercise HR parameters, including HR_aft_, ΔHR, HRR, and HR_max%_. These results are consistent with prior studies demonstrating autonomic dysregulation in smokers, and they provide a genetic explanation for the physiological patterns previously observed. For instance, Papathanasiou et al. [26] showed that young habitual smokers exhibit significantly attenuated heart rate recovery following standardised treadmill exercise, indicative of autonomic impairment. Dinas et al. [27] reviewed active and passive tobacco exposure and reported that smoking reduces heart rate variability and slows recovery dynamics during and after exertion. Barutcu et al. The authors of [12] further confirmed that smokers display heightened sympathetic drive and blunted vagal tone in response to autonomic challenges.

Our findings extend these physiological observations by identifying three smoking-associated SNPs—rs2235186, rs4142041, and rs578776—that may underlie inter-individual differences in autonomic recovery. Each of these variants is linked to genes with plausible roles in cardiovascular regulation.

The rs2235186 variant lies upstream of the *ADRB1* gene, which encodes the β_1_-adrenergic receptor. Altered receptor expression or sensitivity may enhance sympathetic activation during exertion and delay recovery due to sustained catecholamine signalling [28,29]. The rs4142041 variant is associated with *CTNNA3*, a gene involved in dopaminergic reward processing. Although not directly involved in cardiovascular control, central dopaminergic signalling can influence stress sensitivity and parasympathetic withdrawal [30,31], which may contribute to delayed heart rate recovery and increased cardiovascular risk in individuals with heightened dopaminergic reactivity. This interpretation is supported by recent findings demonstrating extensive genetic overlap between smoking behaviour, schizophrenia, and cardiovascular traits, highlighting the role of central nervous system pathways in autonomic modulation [32]. Meanwhile, rs578776 resides within *CHRNA3*, a gene repeatedly linked to nicotine dependence and autonomic modulation via cholinergic pathways [33,34,35,36]. In addition, smoking-induced activation of innate immune pathways—such as the cGAS-STING and Toll-like receptor systems—may amplify sympathetic drive and impair vagal recovery, further contributing to autonomic dysregulation [37].

In contrast, Linneberg et al. [38] reported a significant association between the rs16969968 polymorphism and elevated HR_rest_ in smokers. This result was not replicated in our cohort, possibly due to population-specific factors, environmental influences, or methodological differences in phenotyping. One plausible explanation for the lack of association with HR_rest_ may lie in the metabolic profile of the participants. As HOMA-IR was included as a covariate in our regression models, its role in modulating autonomic balance warrants further consideration. Recent meta-analyses have also identified elevated homocysteine levels—modulated by smoking and genetic polymorphisms—as independent predictors of autonomic dysfunction and endothelial impairment [39]. Insulin resistance is known to enhance sympathetic tone and suppress parasympathetic activity, which may attenuate baseline heart rate variability and obscure genetic associations at rest. In contrast, exercise-induced stress may amplify genotype-dependent differences in autonomic response, revealing stronger SNP associations with post-exercise heart rate traits.

These findings align with known cardiometabolic mechanisms and underscore the value of incorporating metabolic markers into cardiovascular risk assessment. It also highlights the complex interplay between genetic and environmental factors in determining heart rate regulation. One additional reason for the lack of associations with HR_rest_ may be its high variability in response to acute behavioural and environmental stimuli, such as stress, physical activity, caffeine intake, and circadian rhythms [40,41,42]. These short-term modifiers may mask subtle genetic effects. By contrast, exercise-induced heart rate dynamics are governed more directly by intrinsic autonomic processes, such as vagal withdrawal and sympathetic reactivation, and may, therefore, serve as more robust phenotypic markers for detecting genotype-dependent autonomic variation, particularly in populations with metabolic comorbidities.

Recent genome-wide meta-analyses, including data from the UK Biobank and the Global Biobank Meta-analysis Initiative (GBMI), reaffirmed the importance of the *CHRNA3*, *CTNNA3*, and *ADRB1* loci in regulating cardiovascular and autonomic nervous system function. These studies identified robust associations between rs578776 (*CHRNA3*) and nicotine dependence, rs2235186 (*ADRB1*) and heart rate modulation, and other variants across diverse populations [43]. In parallel, advances in Mendelian randomisation methodology—particularly through the MR-Base platform—have enabled more rigorous causal inference using curated GWAS summary statistics and automated sensitivity analyses [44,45]. Integrating these resources strengthens the biological plausibility of our findings and supports the translational potential of genotype-informed cardiovascular risk stratification. Recent Mendelian randomisation meta-analyses have confirmed causal links between genetic liability to smoking and increased risk of atrial fibrillation, coronary artery disease, and autonomic dysfunction across diverse populations [46,47]. Beyond genetic variation, smoking-induced DNA methylation—particularly at loci such as cg25313468 (REST)—has been shown to mediate cardiovascular risk and autonomic imbalance, complementing our SNP-based findings [46]. These epigenetic signals may interact with germline variants to shape individual autonomic profiles, particularly under exertional stress.

Moreover, our three-variant PGS showed robust, graded associations with key exercise heart-rate parameters—ΔHR (*p* = 8.86 × 10^−4^) and HRR (*p* = 5.42 × 10^−6^)—suggesting that genotype-guided risk stratification could identify smokers at greatest risk of autonomic dysfunction. Clinically, carriers at high genetic risk might benefit from targeted β-blocker therapy, nicotinic receptor modulators, or personalised exercise prescriptions designed to enhance vagal reactivation. From a public-health perspective, early genetic screening in tobacco-exposed populations could prioritise cessation support and cardiovascular monitoring for those most predisposed. This interpretation is consistent with the results of emerging Mendelian randomisation studies that support the idea that the genetic architecture associated with smoking plays a causal role in coronary and autonomic dysregulation [15,18].

Several limitations should be acknowledged. First, smoking status was self-reported without biochemical verification (e.g., cotinine levels) or pack-year quantification, which may have led to exposure misclassification. Second, we lacked objective measures of cardiorespiratory fitness—such as direct VO_2_max testing or continuous physical activity monitoring—which could confound heart rate responses. Third, the YMCA 3 min step test relies on a 60 s manual radial pulse count that is vulnerable to inter- and intra-observer variability and may underestimate peak and early recovery values compared with continuous ECG monitoring. Fourth, despite adjusting for a broad range of demographic, metabolic, and lifestyle covariates, residual confounding by unmeasured acute modifiers (e.g., stress, caffeine intake, and circadian influences) and by population stratification or cryptic relatedness may persist. Fifth, our genetic risk score comprised only three SNPs, capturing a limited fraction of the polygenic architecture of autonomic control, and we did not examine potential effect modification by sex or age. Finally, the cross-sectional design, modest sample size (*n* = 661), and high proportion of Roma participants restrict generalisability to other ethnic groups and preclude causal inference. Future research in larger, multi-ethnic, longitudinal cohorts—with continuous ECG monitoring, Mendelian randomisation analyses, and functional studies—will be essential to validate and expand upon these findings.

Emerging research on clonal haematopoiesis suggests that smoking accelerates somatic mutations in hematopoietic stem cells, which may interact with inherited genetic risk loci and influence autonomic recovery. Integrating CHIP markers into future genetic models could enhance cardiovascular risk stratification [48]. Taken together, these multi-layered genomic insights underscore the need for integrative models that combine germline, somatic, and epigenetic data to refine cardiovascular risk prediction in smokers.

## 4. Materials and Methods

### 4.1. Study Design and Populations

A complete and detailed description of the design of this study and the collection of data has been presented in our previous papers [5,49]. In brief, the survey aimed to investigate the underlying factors contributing to the Roma population’s markedly poorer health status compared to the general Hungarian population, particularly with regard to the high burden of non-communicable diseases.

All data were collected in 2018 in two counties of northeastern Hungary (Hajdú-Bihar and Szabolcs-Szatmár-Bereg), which have the highest representation of segregated Roma colonies. This study was conducted by trained healthcare professionals and field workers who followed standardised protocols throughout the data collection process. Ethical approval for the study was granted in 2017 (reference no. 61327-3/2017/EKU).

The Roma sample was drawn from 25 randomly selected segregated colonies, each with over 100 residents. Within each colony, 20 households were randomly selected, and one adult from each household was recruited. The general Hungarian sample consisted of 25 individuals randomly selected from each of 20 general medical practices participating in the General Practitioners’ Morbidity Sentinel Stations Program (GPMSSP) [50].

The survey included adults aged 20–64 years and comprised three core components: (1) a structured questionnaire based on the European Health Interview Survey (EHIS wave 2); (2) standardised physical examinations, including anthropometric and blood pressure measurements; and (3) fasting laboratory testing for basic metabolic and cardiovascular risk markers.

Anthropometric measurements (height, weight, and BMI) were performed using calibrated SECA devices, with participants wearing light clothing and no shoes. Blood pressure was measured using validated Omron M6 Comfort devices, following a standardised protocol: participants were seated and rested for at least five minutes before measurement, and three readings were taken at one-minute intervals. The average of the second and third readings was used for analysis. All equipment was regularly calibrated, and measurements were performed by trained nurses and general practitioners. All measurements were conducted in designated community health centres, ensuring standardised conditions across sites.

Physical activity and health status were assessed using two validated instruments: the European Health Interview Survey (EHIS wave 2) and the International Physical Activity Questionnaire (IPAQ long form). The IPAQ enabled domain-specific and intensity-specific analysis of physical activity, and results were expressed in MET-min/week according to the standardised IPAQ scoring protocol. Both instruments have been widely used in European population studies, and their psychometric properties are well established.

A total of 832 individuals were initially recruited for the study: 417 from the general Hungarian population and 415 from the Roma population. However, participants with incomplete genotype and/or phenotype data were excluded from the final analysis. Specifically, 171 individuals were removed due to missing or invalid data in either the genetic or cardiovascular fitness assessments. As a result, the final analytic sample consisted of 661 individuals (330 non-smokers and 331 smokers), all of whom had complete and validated data for both genotyping and heart rate response measurements. A detailed participant flow diagram illustrating recruitment, exclusions, and final sample size is provided in Appendix A.

### 4.2. DNA Extraction, SNP Selection, Testing Hardy–Weinberg Equilibrium, Linkage Disequilibrium, and Genotyping

DNA was extracted from EDTA-anticoagulated blood samples using the MagNA Pure LC system (Roche Diagnostics, Basel, Switzerland), in accordance with the manufacturer’s instructions.

The process of selecting the SNPs and the results of testing Hardy–Weinberg equilibrium and linkage disequilibrium have been described elsewhere [51]. In brief, a literature search on PubMed was conducted to identify SNPs demonstrating a significant effect on smoking behaviours and that are consistent with smoking intensity. Nine SNPs were selected for inclusion in the present study.

Genotyping was performed on a MassARRAY platform (Sequenom, Inc., San Diego, CA, USA) using iPLEX Gold chemistry by the Mutation Analysis Facility (MAF) service provider at the Karolinska Institute in Solna, Sweden. The Facility conducted validation, concordance analysis, and quality control according to their protocols.

### 4.3. Measurement of Heart Rate Responses to Physical Exertion

The YMCA 3 min step test was used to assess how heart rate changes in response to physical exertion. Each test began with the subject sitting on a chair in a quiet room for a two-minute rest period. They were then instructed to step up and down on a 30 cm step or bench 72 times within three minutes, maintaining a pace set by a metronome at 96 beats per minute (four beats per step cycle, equating to 24 steps per minute).

Heart rate after exercise was recorded following a standardised five-second seated recovery period after completing the step test. This minimised the influence of early autonomic rebound on pulse values. HR was measured at four intervals during the test: one minute before the test, while at rest, immediately after the test, and five minutes (HR_5min_) and ten minutes (HR_10min_) afterwards. Heart rate was assessed manually via radial pulse palpation for 60 s by trained professionals. This method was chosen because it is practical for field-based surveys. Although it may slightly underestimate peak post-exercise values due to rapid early recovery, consistently applying it to all participants introduces a systematic, non-differential error that is unlikely to bias comparative analyses. This field-friendly protocol has been validated in multiple population health surveys and demonstrates excellent reproducibility for both resting and recovery heart-rate measures in large cohorts.

The difference between HR_rest_ and HR_aft_ was defined as a ΔHR, which is inversely correlated with cardiovascular fitness.$$\Delta HR=HR_{aft}-{HR}_{rest}$$

The heart rate recovery coefficient was calculated as an indicator of autonomic cardiovascular regulation [52], expressing the degree of heart rate decline after physical exertion relative to the resting heart rate. It is calculated using the following formula:$$HRR=\frac{{HR}_{aft}-{HR}_{5min}}{{HR}_{rest}}$$

The predicted maximum heart rate [53] and percent of predicted maximum heart rate (HR_max%_) were calculated as follows:$${HR}_{max}=220-age\;(years)$$$${HR}_{max\%}=\frac{{HR}_{aft}}{{HR}_{max}}\times 100$$

### 4.4. Calculation of the Individual Effect of SNPs and the Joint Effect Estimated by Genetic Risk Score

To evaluate the effect of each SNP individually, we tested the three most common inheritance models—dominant, recessive, and codominant—and selected the model that exhibited the strongest association with ΔHR. The selection of the best-fitting heritability model was guided by the R^2^ value (higher values indicate a better fit) and *p*-values (lower values indicate a stronger association) [54]. The most appropriate inheritance model for each SNP—based on its association with ΔHR—was then used to construct the genetic risk score (GRS).

SNP coding followed the criteria of the selected inheritance models:(a)Codominant model: the homozygous genotype with the risk allele was coded as 2, the heterozygote as 1, and the homozygous genotype with no risk allele as 0.(b)Dominant model: genotypes with one or two risk alleles were coded as 2; those with no risk allele were coded as 0.(c)Recessive model: genotypes with two risk alleles were coded as 2; both the heterozygote and the homozygous genotype with no risk allele were coded as 0.

To explore the genetic predisposition to smoking-related alterations in cardiovascular physiology, particularly changes in heart rate (ΔHR), a genetic risk score (GRS) was calculated. SNPs previously associated with smoking behaviour and nicotine dependence were examined using multivariable linear regression models. The optimisation process started with the SNP showing the strongest association with ΔHR (lowest *p*-value) and then included additional SNPs in ascending order of *p*-values. SNPs were only retained in the final oGRS if they strengthened the model’s association with ΔHR. Variants that failed to enhance or weakened the predictive value were excluded. The final oGRS was then used to stratify individuals into tertiles representing low, moderate, and high genetic susceptibility to smoking-induced heart rate changes.

For further analyses to minimise the risk of type I error inflation due to multiple testing, all SNPs were analysed simultaneously within a single statistical model, rendering multiple comparison correction (e.g., *p*-value adjustment) unnecessary. However, for statistical calculations with GRS, a Bonferroni corrected *p*-value was considered significant (*p* < 0.016).

### 4.5. Statistical Analyses

Categorical variables were compared using the chi-squared test, while comparisons between two subgroups were conducted using the Mann–Whitney U test. The Kolmogorov–Smirnov test was used to check whether the quantitative variables were normally distributed. Where this was not the case, Templeton’s two-step method [55] was used to transform the variables into a normal distribution. The Jonckheere–Terpstra trend test was used to analyse the association trend between GRS categories and HR-related parameters.

Linear regression analyses were performed, adjusting for the following covariates: age, waist circumference, BMI, systolic and diastolic blood pressure, the HOMA-IR index, the domains of physical activity (work-related, transport-related, domestic, and leisure-time), sitting time, the lipid profile (LDL-C, HDL-C, and triglycerides), the exercise-related polygenic score [56], sex, self-identified Roma ethnicity, financial status, educational level, alcohol consumption, and the use of antihypertensive, antidiabetic, and lipid-lowering medications.

Statistical analyses were conducted using the IBM Statistical Package for the Social Sciences (SPSS) version 26 (Armonk, NY, USA).

### 4.6. Ethical Approval

The protocol (61327-2017/EKU) was approved by the Committee of the Hungarian Scientific Council on Health. All subjects agreed to participate in the study by signing a written informed consent form.

## 5. Conclusions

Our findings suggest that smoking-related genetic polymorphisms may not influence HR_rest_ but can significantly modulate acute cardiovascular responses to physical exertion. These genotype-dependent differences in post-exercise HR regulation highlight the importance of context-dependent genetic effects, specifically, how genetic predisposition manifests under physiological stress rather than at rest [42]. Given the established prognostic relevance of abnormal heart rate recovery [57,58], such genetically modulated responses may serve as early indicators of cardiovascular autonomic dysfunction. The strong predictive value of the GRS underscores the potential utility of genetic screening in improving cardiovascular risk stratification, particularly in individuals with a high level of tobacco exposure. As public health moves towards more personalised prevention, genotype-informed models may guide more precise lifestyle recommendations, early interventions, and resource allocation in at-risk populations.

## Figures and Tables

**Figure 1 ijms-26-08787-f001:**
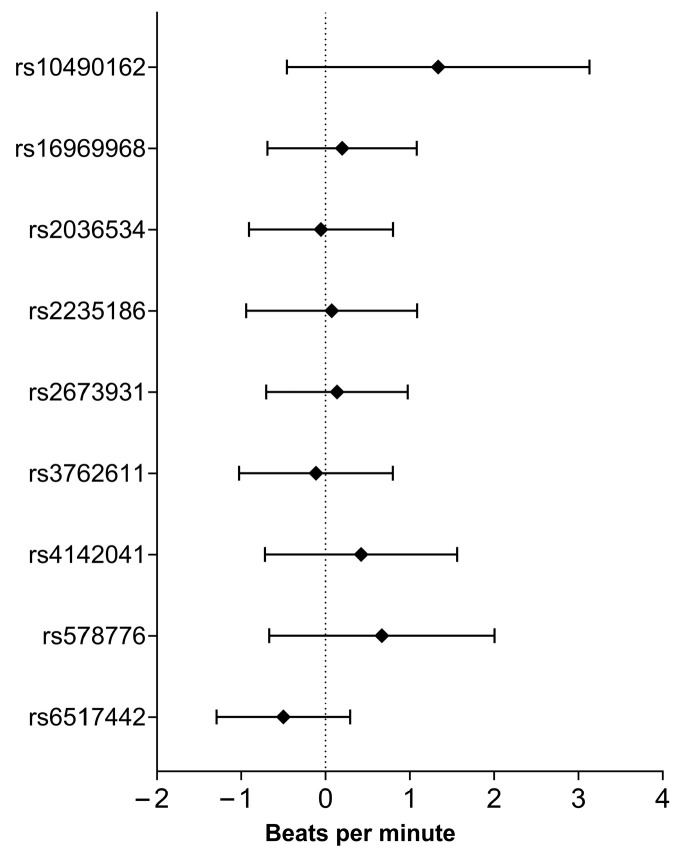
Smoking-related SNP association with resting heart rate based on adjusted linear regression analysis.

**Figure 2 ijms-26-08787-f002:**
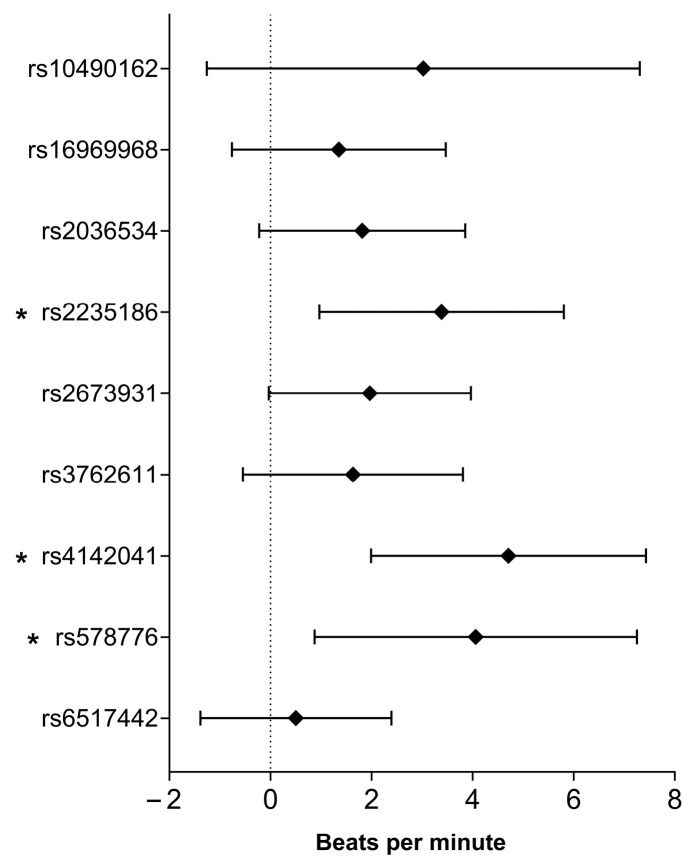
Smoking-related SNPs association with heart rate after exercise (HR_aft_) based on adjusted linear regression analysis. *: *p* < 0.05.

**Figure 3 ijms-26-08787-f003:**
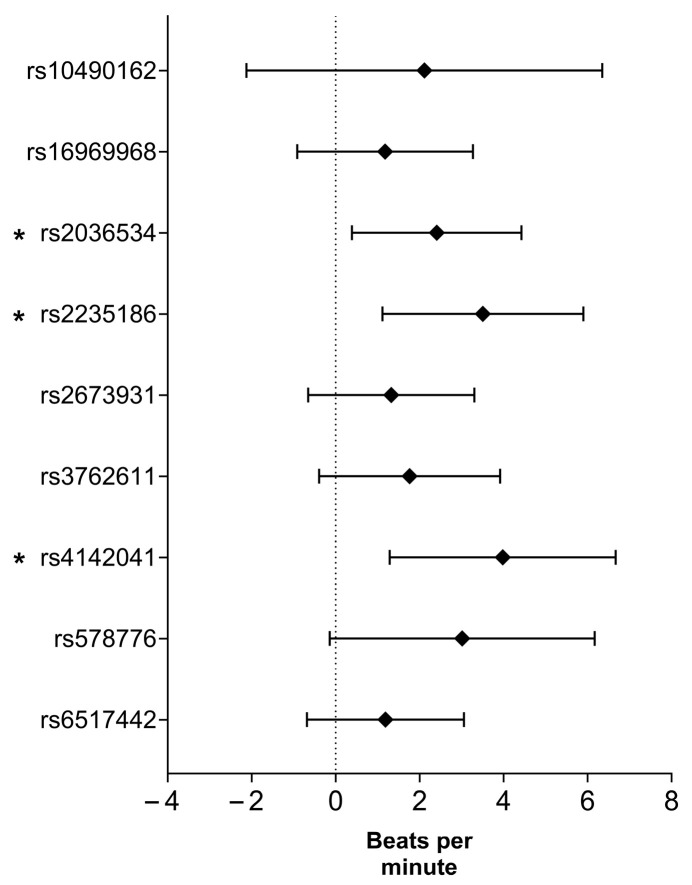
Smoking-related SNPs association with delta heart rate based on adjusted linear regression analysis. *: *p* < 0.05.

**Figure 4 ijms-26-08787-f004:**
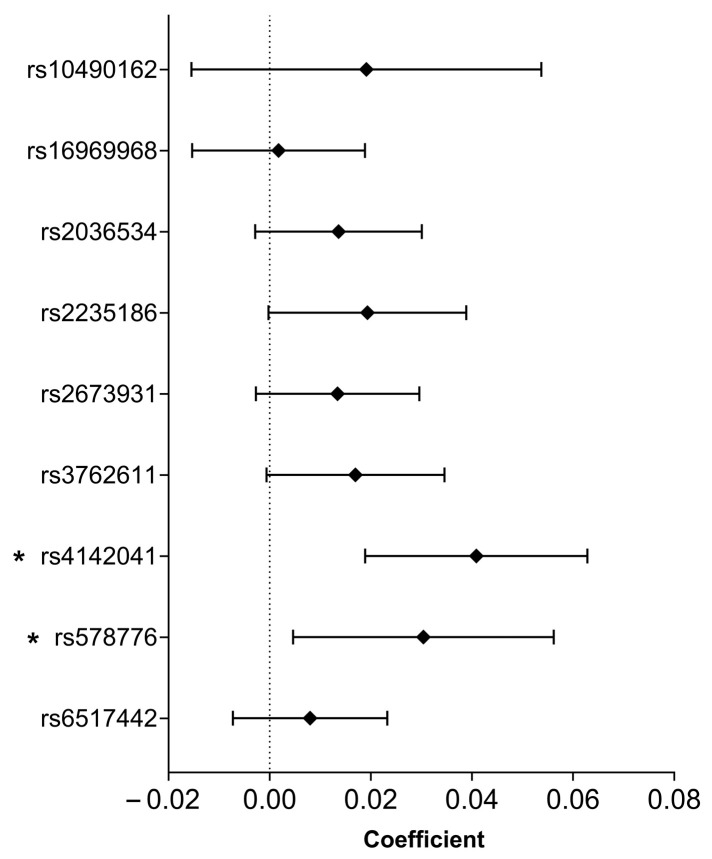
Smoking-related SNPs association with the heart rate recovery coefficient based on adjusted linear regression analysis. *: *p* <0.05.

**Figure 5 ijms-26-08787-f005:**
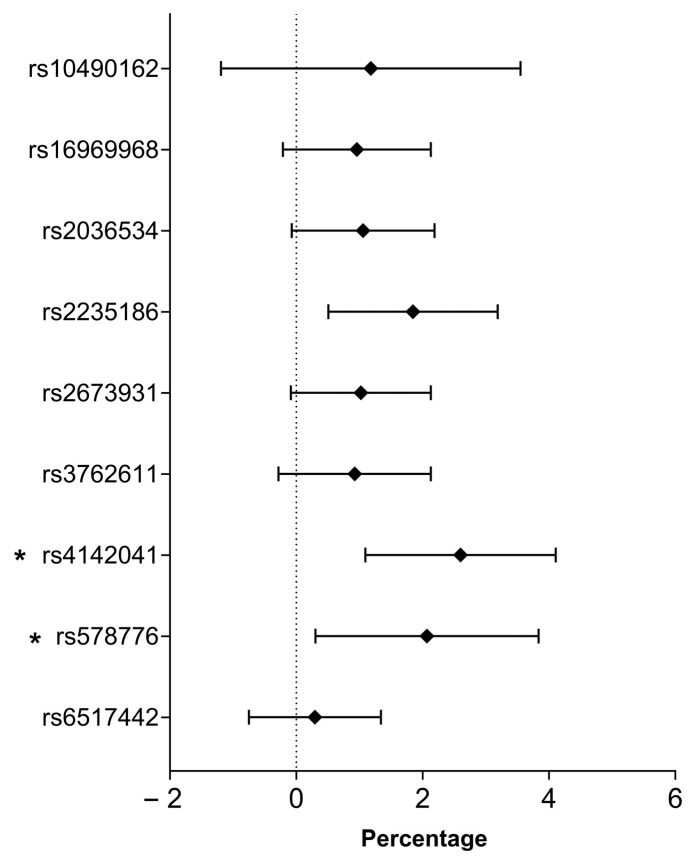
Smoking-related SNPs association with a percentage of predicted maximum heart rate based on adjusted linear regression analysis. *: *p* <0.05.

**Table 1 ijms-26-08787-t001:** Baseline characteristics of non-smokers and smokers. Includes anthropometric, metabolic, physical activity, cardiovascular, and sociodemographic variables.

		Non-Smokers (*n* = 330)	Smokers (*n* = 331)	*p*-Value
		Average (95% CI)		
Age (years)		43.68 (42.29–45.08)	42.49 (41.16–43.82)	0.182
Waist circumference (cm)		97.22 (95.55–98.88)	92.71 (90.96–94.45)	<0.001 *
BMI (kg/m^2^)		27.96 (27.32–28.59)	26.51 (25.83–27.19)	0.001 *
Systolic blood pressure (mmHg)		126.24 (124.67–127.82)	124.24 (122.26–126.21)	0.048 *
Diastolic blood pressure (mmHg)		79.90 (78.99–80.82)	78.62 (77.48–79.76)	0.035 *
Homa-IR		4.37 (3.61–5.14)	3.70 (3.10–4.31)	0.046 *
Domains of physical activity	Work (MET-min/week)	4881.90 (4229.64–5534.16)	4924.11 (4251.23–5596.99)	0.516
	Transport (MET-min/week)	1345.26 (1153.32–1537.20)	1669.42 (1430.87–1907.98)	0.007 *
	Domestic (MET-min/week)	2930.61 (2616.53–3244.68)	2976.23 (2654.54–3297.93)	0.790
	Leisure-time (MET-min/week)	1340.36 (1137.50–1543.22)	1007.11 (819.53–1194.69)	<0.001 *
Sitting time (min/week)		529.29 (479.46–579.11)	413.44 (383.16–443.73)	0.001 *
Low-density lipoprotein cholesterol (mmol/L)		3.11 (3.01–3.21)	3.15 (3.04–3.25)	0.517
Triglycerides (mmol/L)		1.56 (1.44–1.67)	1.55 (1.44–1.66)	0.750
High-density lipoprotein cholesterol (mmol/L)		1.37 (1.33–1.42)	1.26 (1.22–1.30)	<0.001 *
Resting heart rate (bpm)		77.20 (76.13–78.27)	77.61 (76.43–78.79)	0.808
Heart rate after exercise (bpm)		109.79 (107.38–112.21)	112.25 (108.94 115.55)	0.378
Delta heart rate (bpm)		32.59 (30.30–34.88)	34.64 (31.45–37.82)	0.171
Heart rate after 5 min (bpm)		91.92 (90.24–93.60)	94.55 (92.59–96.51)	0.332
Heart rate after 10 min (bpm)		81.13 (79.95–82.31)	82.85 (81.44–84.25)	0.213
Heart rate recovery coefficient		0.23 (0.21–0.26)	0.23 (0.21–0.25)	0.338
Maximum heart rate expressed as percentage		62.60 (61.15–64.05)	63.46 (61.57–65.36)	0.446
		Average prevalence in % (95% CI)		*p*-value
Women		63.03 (57.73–68.11)	68.81 (63.51–73.77)	0.123
Roma		32.12 (27.26–37.30)	69.13 (63.84–74.07)	<0.001 *
Financial status	Bad	14.85 (11.33–18.98)	25.72 (21.11–30.79)	<0.001 *
	Average	55.15 (49.76–60.45)	56.59 (51.04–62.02)	
	Good	30.00 (25.25–35.10)	17.68 (13.75–22.21)	
Education	Less than primary and primary	36.36 (31.31–41.65)	71.70 (66.51–76.49)	<0.001 *
	Vocational and high school	48.79 (43.43–54.17)	24.76 (20.21–29.77)	
	College and university	14.85 (11.33–18.98)	3.54 (1.89–6.04)	
Alcohol consumption	Less than 1 time per month	49.09 (43.72–54.47)	50.80 (45.26–56.33)	0.641
	1 time per month	33.03 (28.12–38.24)	34.08 (28.98–39.48)	
	More than 2 times per month	17.88 (14.03–22.28)	15.11 (11.46–19.41)	
Anti-hypertensive medication		31.82 (26.97–36.99)	26.37 (21.70–31.47)	0.129
Anti-diabetic medication		7.27 (4.84–10.45)	9.00 (6.20–12.56)	0.423
Lipid-lowering medication		9.39 (6.60–12.90)	9.32 (6.47–12.93)	0.976

BMI: body mass index; HOMA-IR: Homeostatic Model Assessment of Insulin Resistance; MET-min/week: Metabolic Equivalent of Task minutes per week; bpm: beats per minute; HDL-C: high-density lipoprotein cholesterol; LDL-C = low-density lipoprotein cholesterol. *: Statistically significant difference (*p* < 0.05).

**Table 2 ijms-26-08787-t002:** Effects of smoking-related SNPs on delta heart rate using a dominant, codominant, and recessive inheritance model.

SNP (Risk Allele)	Inheritance Model	B (95% CI)	*p*-Value	R^2^
rs10490162 (C)	Recessive	0.542 (−7.295–8.379)	0.892	0.174
	**Codominant**	**1.820 (−2.426–6.065)**	**0.370**	**0.175**
	Dominant	1.080 (−1.285–3.446)	0.400	0.175
rs16969968 (A)	Recessive	0.797 (−2.102–3.696)	0.589	0.175
	Codominant	1.833 (−0.947–4.613)	0.196	0.176
	**Dominant**	**1.340 (−0.540–3.219)**	**0.162**	**0.177**
rs2036534 (C)	Recessive	0.808 (−3.255–4.871)	0.696	0.174
	Codominant	2.600 (−0.518–5.717)	0.102	0.178
	**Dominant**	**1.765 (−** **0** **.140–3.670)**	**0.069**	**0.179**
rs2235186 (G)	Recessive	1.522 (−0.393–3.437)	0.119	0.177
	Codominant	3.241 (0.830–5.652)	0.008	0.184
	**Dominant**	**4.059 (1.654–6.464)**	**9.74 × 10^−4^**	**0.189**
rs2673931 (T)	**Recessive**	**1.198 (−0.805–3.200**	**0.236**	**0.176**
	Codominant	1.650 (−1.084–4.385)	0.241	0.176
	Dominant	0.870 (−1.602–3.341)	0.490	0.175
rs3762611 (G)	**Recessive**	**2.408 (0.252–4.565)**	**0.029**	**0.181**
	Codominant	3.479 (−0.134–7.092	0.059	0.179
	Dominant	0.441 (−4.798–5.679)	0.869	0.174
rs4142041 (A)	Recessive	0.497 (−1.410–2.404)	0.609	0.175
	Codominant	2.427 (−0.301–5.155)	0.081	0.178
	**Dominant**	**3.825 (1.110–6.541)**	**0.006**	**0.185**
rs578776 (G)	Recessive	−0.019 (−1.897–1.859)	0.984	0.174
	Codominant	1.229 (−1.460–3.919)	0.370	0.175
	**Dominant**	**2.661 (−** **0** **.109–5.431)**	**0.060**	**0.179**
rs6517442 (C)	Recessive	1.265 (−1.836–4.366)	0.423	0.175
	Codominant	2.761 (−0.070–5.592)	0.056	0.179
	**Dominant**	**1.948 (** **0** **.079–3.817)**	**0.041**	**0.180**

Heritable models with the strongest correlation for SNPs are highlighted in bold.

**Table 3 ijms-26-08787-t003:** Trend analysis of heart rate-related parameters across genetic risk score categories.

	Genetic Risk Score			*p* for Trend
	0–2 (*n* = 41)	4 (*n* = 219)	6 (*n* = 381)	
	Average (95% CI)			
HR_rest_	75.32 (72.69–77.95)	77.03 (75.63–78.42)	77.84 (76.81–78.87)	0.253
HR_aft_	98.83 (91.72–105.94)	105.75 (102.75–108.76)	115.30 (112.53–118.07)	1.47 × 10^−8^ **
ΔHR	23.51 (16.87–30.16)	28.73 (26.02–31.43)	37.46 (34.74–40.18)	8.12 × 10^−7^ **
HRR	0.14 (0.10–0.19)	0.19 (0.17–0.22)	0.26 (0.24–0.29)	6.19 × 10^−8^ **
HR_max%_	55.74 (51.77–59.72)	60.20 (58.37–62.02)	65.42 (63.83–67.02)	6.34 × 10^−8^ **

HR_rest_: resting heart rate; HR_aft_: heart rate after the test; ΔHR: delta heart rate; HRR: heart rate recovery coefficient; HR_max%_: percent of predicted maximum heart rate; **: *p* < 0.016 (Bonferroni corrected *p*-value).

**Table 4 ijms-26-08787-t004:** Association of genetic risk score on heart rate-related parameters.

	B (95% CI)	*p*-Value
HR_rest_	0.301 (−0.322–0.925)	0.343
HR_aft_	3.986 (2.486–5.486)	2.47 × 10^−7^ **
ΔHR	1.640 (0.676–2.604)	8.86 × 10^−4^ **
HRR	0.028 (0.016–0.040)	5.42 × 10^−6^ **
HR_max%_	2.193 (1.362–3.023)	2.93 × 10^−7^ **

HR_rest_: resting heart rate; HR_aft_: heart rate after the test; ΔHR: delta heart rate; HRR: heart rate recovery coefficient; HR_max%_: percent of predicted maximum heart rate, **: *p* < 0.016 (Bonferroni corrected *p*-value).

## Data Availability

Due to data protection and ethical concerns, the dataset(s) supporting the conclusions of this article are available upon request from the study coordinators, Prof. Róza Ádány (adany.roza@med.unideb.hu) and Dr. Péter Pikó (piko.peter@med.unideb.hu).

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
