# Peer review of "The Impact of Smoking-Associated Genetic Variants on Post-Exercise Heart Rate"

_ijms, 2025, doi:10.3390/ijms26188787_

Round 1
Reviewer 1 Report
Comments and Suggestions for Authors
The Impact of Smoking-Associated Genetic Variants on Post- Exercise Heart Rate
Reviewer Comments – Major Revision Required
Thank you for the opportunity to review this manuscript. The research topic is relevant and timely, particularly as it explores the genetic basis of cardiovascular response to exercise among smokers. However, the current version of the manuscript requires major revisions in terms of structure, clarity, methodological transparency, and consistency. The authors are encouraged to address the following issues in detail:
- Manuscript Structure – Reorder Required
The manuscript does not currently follow the standard format for original research articles. Please revise the structure to the following sequence:
Introduction
Methods
Results
Discussion
Conclusion
This structure ensures scientific clarity, flow, and reader comprehension.
- Methods Section – Lacks Critical Detail
The Methods section requires substantial elaboration. Specific concerns include:
Study Duration and Setting: Clearly state when and where the study was conducted.
Participant Recruitment: Describe the recruitment process, population source (general or Romanian), and inclusion/exclusion criteria.
Anthropometric and Clinical Data Collection:
Detail how blood pressure, height, weight, and BMI were measured.
Specify the instruments used, the protocols followed (e.g., number of BP readings, resting time), and whether equipment was calibrated.
Identify who performed the measurements (e.g., trained personnel, clinicians).
Use of Questionnaires or Tools: If any validated instruments or questionnaires were used, provide their names and appropriate references.
Without these details, the study’s validity, reproducibility, and scientific rigor cannot be evaluated appropriately.
- Inconsistencies in Participant Numbers
There is a clear discrepancy between the Methodology and Results sections:
The Methods section states that 832 participants were included.
However, the Results section and tables reflect data for only around 600+ participants, with no explanation provided.
The authors must clearly account for this difference by answering the following:
How many participants were initially recruited?
How many were excluded, and on what basis?
How many were lost to follow-up or did not meet inclusion criteria?
Were any participants removed due to missing or incomplete data?
These details should be explicitly stated in the Methods section and ideally illustrated in a participant flow diagram.
- Results Section – Requires Clear Structure
The Results section should begin with:
The final number of participants included in the analysis.
A detailed summary of baseline characteristics (age, gender, smoking status, etc.).
Additionally:
Tables should be clearly labeled, correspond directly with the text, and be self-explanatory.
Ensure consistency between the data presented in the narrative and the tables.
- Discussion Section – Needs Depth and References
The Discussion should:
Compare the findings with those of previous studies.
Discuss the potential mechanisms underlying observed associations.
Outline the implications of the results.
End with a Limitations paragraph clearly stating any study weaknesses or constraints.
Moreover, references throughout the manuscript—especially in the Introduction, Methods, and Discussion—are insufficient or missing. Please support your study with relevant and up-to-date literature.
- Previously Raised Comments Not Addressed
It appears that many of the issues highlighted in a prior round of review (including structure, participant accounting, and methodology clarity) have not been sufficiently addressed. It is essential that the authors respond point-by-point to all reviewer comments in their revised submission.
Recommendation:
Major Revision Required
This study has potential, but it requires substantial restructuring and improvement in methodological transparency and data consistency. Authors are strongly encouraged to revise the manuscript thoroughly and provide detailed responses to all reviewer comments.
Reviewer 2 Report
Comments and Suggestions for Authors
Main Findings
Three smoking-related SNPs (rs2235186 – ADRB1, rs4142041 – CTNNA3, rs578776 – CHRNA3) were significantly associated with post-exercise HR dynamics (HRaft, ΔHR, HRR, HRmax%), but not with resting HR.
Genetic effects appear stronger on recovery-related cardiovascular traits than on baseline heart rate, likely due to reduced influence of environmental/behavioral noise.
Cumulative genetic risk score based on these SNPs robustly predicted exercise-induced HR responses.
ADRB1 variant (rs2235186) → may alter β₁-adrenergic receptor sensitivity, increasing sympathetic drive.
CTNNA3 variant (rs4142041) → may affect dopaminergic reward pathways, influencing autonomic stress reactivity.
CHRNA3 variant (rs578776) → linked to nicotine dependence and altered cholinergic signaling, impacting HR regulation. Together, these loci plausibly modulate autonomic efficiency during stress and recovery.
1-Abstract
- The abstract is comprehensive but comes across as overly detailed and complex. Consider streamlining by focusing on the most important SNPs and avoiding listing all rs-numbers.
- The level of methodological detail (e.g., full test protocol and multiple HR indices) may overwhelm readers. You could summarize key measures and move technical specifics to Methods.
- The statistical reporting is precise, but extensive use of coefficients and p-values in the abstract reduces readability. Highlight significant findings in plain terms.
- Structuring the abstract more clearly into background, aim, methods, results, and conclusion would improve flow and make the key message easier to grasp.
- Overall, the results are interesting and relevant, but the abstract could be made leaner and more impactful with tighter wording and emphasis on the main discoveries.
2- Introduction
The introduction is comprehensive and cites a wide range of relevant references, but the material would benefit from clearer organization to improve readability and flow.
3- Discussion
The discussion is comprehensive and well-grounded; however, it would be strengthened by incorporating insulin resistance (HOMA-IR) into the interpretation of your findings. Both study groups showed elevated HOMA-IR, and insulin resistance is known to influence autonomic balance, typically increasing sympathetic activity and reducing vagal tone. This could help explain why baseline HR (HRrest) associations were absent, while post-exercise heart rate traits showed robust SNP associations.
We therefore recommend that you briefly discuss how HOMA-IR might act as a potential confounder or effect modifier in the relationship between smoking-related SNPs and heart rate dynamics. Highlighting this angle would add biological depth and align your genetic findings with well-established cardiometabolic pathways, thereby increasing the translational weight of the manuscript.
Round 2
Reviewer 1 Report
Comments and Suggestions for Authors
All comments have been satisfactorily addressed.
Reviewer 2 Report
Comments and Suggestions for Authors
The authors have undertaken major revisions and have thoroughly addressed the reviewers’ comments. As a result, the manuscript has been considerably strengthened and is now in a much-improved form.